# A Scalable Technique for Weak-Supervised Learning with Domain Constraints

**Sudhir Agarwal   Anu Sreepathy   Lalla Mouatadid**
Intuit AI Research Center, Mountain View, CA, USA
`{firstname_lastname@intuit.com}`

## Abstract

We propose a novel scalable end-to-end pipeline that uses symbolic domain knowledge as constraints for learning a neural network for classifying unlabeled data in a weak-supervised manner. Our approach is particularly well-suited for settings where the data consists of distinct groups (classes) that lends itself to clustering-friendly representation learning and the domain constraints can be reformulated for use of efficient mathematical optimization techniques by considering multiple training examples at once. We evaluate our approach on a variant of the MNIST image classification problem where a training example consists of image sequences and the sum of the numbers represented by the sequences – e.g., (⊡, ⊡, 9), and show that our approach scales significantly better than previous approaches that rely on computing all constraint satisfying combinations for each training example.

## 1   Introduction

Integrating logical reasoning and machine learning is one of the main challenges of AI. Doing so can lead to systems that can solve more complex problems, learn from less data, comply with domain knowledge, or have better performance [1, 2].

In particular, incorporating domain knowledge as symbolic constraints during ML training has shown to improve the accuracy of ML predictions by learning an ML model that tries to enforce the constraints as much as possible. These techniques are especially promising when training data is limited and constraints are available or can be easily acquired. Some recent approaches handle constraints by incorporating probabilities from neural networks into logic programs. For instance, models that incorporate constraints expressed in ProbLog [3] and Answer Set Programs have been proposed in [4] and [5] respectively. Some other approaches such as [6] and [7] use stochastic grammar based logic programs and probabilistic deductive database with differentiable reasoning respectively. However, due to the limitations arising from model grounding, all these approaches don't scale well when the complexity of the problem increases. The grounding of a constraint is the computation of all satisfying assignments of the constrained variables, which can lead to intractable combinatorial blow-up in cases with a large number of constrained variables. This problem, known as the grounding bottleneck, arises in logical reasoning domains, such as statistical relational AI (StarAI) [8] and Answer Set Programming (ASP) [9]. While the techniques presented in [4, 5, 6] are highly expressive and accurate, they rely on naively iterating through all possible output combinations for each training example in order to compute the satisfying output combinations. For instance, for (⊡, ⊡, 5), they would iterate over $10^2$ combinations in order to compute the satisfying combinations $(0, 5), (1, 4), (2, 3), (3, 2), (4, 1), (5, 0)$. While expressiveness can be traded for scalability – DeepStochLog [6] vs. DeepProbLog [4] –, the combinatorial blow-up still poses a major hurdle for more complex problems; with $n$ images per example, the number of possible output combinations grows exponentially ($10^n$). To the best of our knowledge, there is no previous approach that can be used for a multitude of industry use cases, which often have complex domain constraints and require high scalability.

Has it Trained Yet? Workshop at the Conference on Neural Information Processing Systems (NeurIPS 2022).

In this work, we develop a scalable weak-supervised learning technique to incorporate symbolic domain constraints into neural networks. Our key contribution is an end-to-end pipeline that avoids the grounding bottleneck and scales to state-of-the-art results on a challenging neuro-symbolic problem. Our approach is based on the insight that in some cases (a) the data consists of distinct groups (e.g., digit images) that are partially recoverable through clustering-friendly representation learning, and (b) the constraint (e.g., addition) is the same across all training examples and the properties of the constraint are known and can be exploited to our advantage.

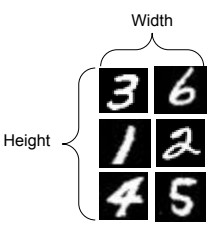

In this paper, we focus on a variant of the MNIST image classification problem where each training example is, instead of an (image, label) pair, a sequence of $w \times h$ MNIST images and an integer $s$ where $w$ (*width*) represents the number of digits in a number and $h$ (*height*) represents the number of numbers in the example whose sum is equal to $s$. Unlike in the standard MNIST classification problem, since we do not have the image labels, we first infer the labels, and then use them to train the classifier. A practical example of such a use case is in information extraction from hand-filled forms such as financial documents (e.g., loan and tax forms) where certain "total" fields are pre-filled in typesetting, while other fields leading up to the total are handwritten information from the user.

Figure 1: An example with $h = 3, w = 2$, and $s = 93$.

For the MNIST classification variant, the state-of-the art solution [6] has been shown to work for $w \leq 4, h = 2$. We show that our approach scales in both width and height as our model achieves between 92% and 97% accuracy for $w \leq 10, h \leq 6$ with the training time being independent of $w$ and $h$.

## 2   Method

Overall, our solution can be broken down into the four main steps as presented in Figure 2.

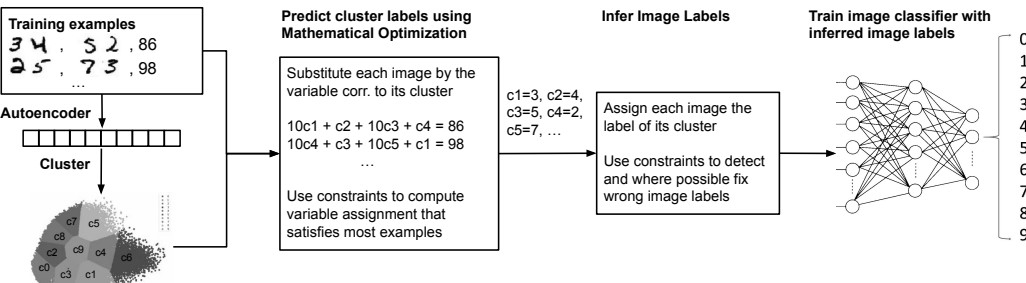

Figure 2: The main steps of our proposed method.

**1. Autoencoder-based clustering**: We first pre-train a fully connected symmetric autoencoder with dense layer dimensions $[500, 500, 2000, 10]$ for 300 epochs, similar to the one in [10], then use the weights of the encoder to cluster with k-means using k-means++ initialization and 10 clusters. The input for this step is the set of individual images across all training examples.

**2. Cluster label prediction**: A training example of width $w$, height $h$, and sum $s$ can be represented as a linear equation $\sum_{i=1}^{h} \sum_{j=1}^{w} v(img_{i,j}) \times 10^{w-j} = s$, where $v(img_{i,j})$ represents the variable assigned to the cluster that contains the $j^{th}$ image in the $i^{th}$ number of the training example. If the clustering was 100% accurate, one could simply assign each cluster $c_i, i \in [0, 9]$ a variable $v_i$, and solve a system of linear equations to determine the values of each $v_i$. However, since we don't start with a clustering of 100% purity, in this step, we formulate the problem as an integer linear program using *CVXPY* [11, 12]. We split the training examples into batches of size 100, solve the optimization problem for each batch using $L^1$-norm as the objective function, and determine the overall result as the batch result that satisfies the most training examples over all batches. As a result, this step assigns each cluster an integer from $[0, 9]$.

**3. Image label inference**: The cluster labels computed in the previous step give the initial labeling for each image in the training dataset. This labeling can be at most as accurate as the clustering purity. The goal of this step is to improve image labeling based on rule-based inference. Currently, this step consists of only one rule based on the fact that a variable in a linear equation can be resolved if all other variables are known. We use an iterative process by which we increasingly resolve variables in the system of equations with every iteration. In order to be able to do so, we need a way to detect correctly clustered images. Since we do not know with certainty which images are correctly clustered, we assume the images closer to their respective cluster's centroid as correctly labeled as they have a higher chance of being correctly clustered.

---

**Algorithm 1** Image label inference

---

1: $Correct \leftarrow \{\}$
2: **for** $1 \leq radius \leq 5$ **do**
3:      $NewCorrect \leftarrow GetImagesWithinRadius(radius)$
4:      $Correct \leftarrow Correct \cup NewCorrect$
5:      $Changed \leftarrow True$
6:      **while** $Changed = True$ **do**
7:          $Changed \leftarrow InferCorrectLabels(Correct)$
8:      **end while**
9: **end for**

---

**Algorithm 2** InferCorrectLabels

---

1: $Changed \leftarrow False$
2: **for all** $Ex \in TrainingExamples$ **do**
3:      $UnresolvedImages \leftarrow GetUnresolvedImages(Correct, Ex)$
4:      **if** $|UnresolvedImages| = 1$ **then**
5:          Let $UnresolvedImages = \{img\}$
6:          $ResolveImageLabel(Correct, Ex, img)$
7:          $Correct \leftarrow Correct \cup \{img\}$
8:          $Changed \leftarrow True$
9:      **end if**
10: **end for**
11: **return** $Changed$

---

The image label inference (Algorithm 1) works as follows. It starts with an empty set of correctly clustered images. At every iteration $1 \leq radius \leq 5$, it gets the images that are at most $radius$ away from their resp. cluster centroids using the method *GetImagesWithinRadius*, and adds them to the set of correctly labeled images. Based on this set of correctly labeled images, the algorithm then tries to infer further (correct) image labels. It does so by using the method *InferCorrectLabels* (Algorithm 2) in a while loop as long as it can infer new image labels as correct labels. *InferCorrectLabels* iterates over all training examples. *Ex* denotes the training example considered in one such iteration. *InferCorrectLabels* first computes the set of unresolved images in *Ex*, i.e., the images in *Ex* that have not been identified as correct thus far (Line 3). It does so by calling the *GetUnresolvedImages* method, which simply iterates over the images in the given training example, and returns the ones that are not in the set of correctly labeled images thus far. If a training example has only one unresolved image, say *img*, (Line 4), then *InferCorrectLabels* calls the method *ResolveImageLabel* to compute its correct label (Line 6). *ResolveImageLabel* then computes the correct label by substituting all images in *Ex* except *img* by their respective labels and solving a linear equation for *img* such that the sum of the numbers represented by the sequences of images in *Ex* is equal to the sum given in *Ex*. Finally, *ResolveImageLabel* updates the label of *img* to the inferred label. As last step of the iteration, *InferCorrectLabels* adds *img* to the set of correct images (Line 7).

**4. Classification**: The final step of the pipeline is to train a CNN-based classifier [13] using the final inferred image labels from Step 3. The network consists of a convolutional layer with 32 filters and a kernel of size $3 \times 3$, followed by a max pooling layer, two more convolutional layers with 64 filters and kernel size $3 \times 3$ each, another max pooling layer and a dense layer of 100 nodes before the output layer. All layers use ReLu activation and He weight initialization. We use stochastic gradient descent optimizer with a learning rate of 0.01 and momentum of 0.9. We train for 10 epochs.

## 3 Results

Figures 3 and 4 show the classification accuracy and training time respectively for varying widths and heights. For every $w \times h$ combination, the training dataset uses all 60K images in the MNIST training dataset exactly once. This means that the higher the $w \times h$ combinations, the smaller the dataset. Yet our accuracy remains above $90\%$. The reported accuracy is the accuracy of the MNIST classifier (Step 4) using the inferred image labels (Step 3) as described in Section 2. The reported time is the total time for all 4 steps of the pipeline [1]. All our experiments are run on a MacBook Pro 2021 (Apple M1 Max, 64GB).

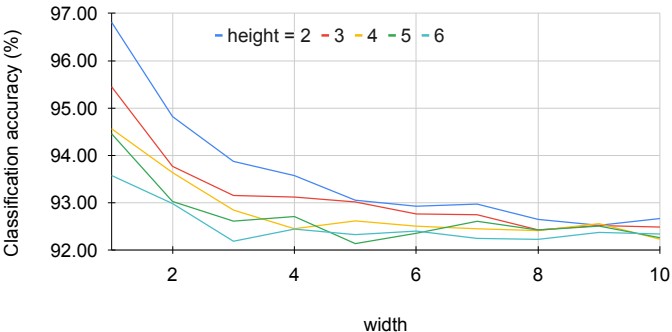

Figure 3: Classification accuracy % for varying $w \times h$ combinations.

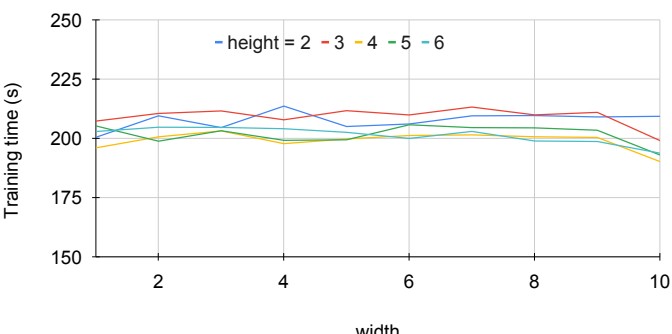

Figure 4: Total training time in seconds for varying $w \times h$ combinations.

As previously mentioned, DeepStochLog [6], the state-of-the-art solution so far, can solve for examples with $h = 2, w \leq 4$, and scales better than earlier approaches [4, 5], both of which can handle $h = 2, w \leq 2$. As shown by the results in Figure 3, our approach not only scales horizontally ($w > 4$), but also vertically ($h > 2$). Furthermore, as shown in Figure 4 our approach doesn't blow up in the total training time as $w$ and $h$ grow. Our clustering component (Step 1), has about $91.3\%$ purity. As a consequence, after the optimization step (Step 2), assuming the optimizer finds the correct cluster labels, the image label accuracy is also about $91.3\%$, which leads to the classification accuracy between $92\%$ and $93\%$. The reason for higher classification accuracy for smaller $w, h$ combinations is that in these cases the image label inference algorithm (Step 3) has a higher chance of inferring the correct image label since it is more likely that an example has only one unresolved image that can be resolved. This increases the accuracy of the final image labels used for training the classifier (Step 4).

In order to compare the addition accuracy with that of DeepStochLog [6], we augmented our pipeline with a component for performing addition on the output of our classifier. With this architecture, we get an addition accuracy of $95\%$, $87\%$, $78.5\%$ and $72\%$ respectively for $w = 1$ to $w = 4$ and $h = 2$. The

---

[1]The total training time we report doesn't include the autoencoder training time. This step takes roughly 30 minutes. We trained the autoencoder once and used the same encoder weights for all $w \times h$ runs since the input to the autoencoder is just the set of individual images across all training examples and is independent of $w$ and $h$.

reason for lower addition accuracy than the classification accuracy (Figure 3) is that with increasing number of images per example it is more likely that a training example has at least one wrongly classified image resulting in a wrong prediction of the sum. With over 99% classification accuracy, we would get addition accuracy similar to that of DeepStochLog. Improving the classification accuracy is a planned future work as discussed in Section 4.

One reason why we can avoid the combinatorial blow-up is attributed to the nature of the problem. The addition constraint is linear and can be solved efficiently for a set of examples if they can be formulated using the same set of variables. In our case, this is made possible by the clustering step which essentially reduces the number of variables from 60k (one for each image) to 10. Furthermore, our approach illustrates the trade-off between expressiveness and the scalability of the language. Similar to how DeepStochLog scales better than DeepProbLog by using stochastic logic programs that are less expressive than probabilistic logic programs used by DeepProbLog, our approach scales even further but can only work for problems that fit our "cluster-then-optimize" paradigm.

## 4  Discussion & Future Work

Our classification accuracy depends strongly on the clustering purity since the image label accuracy after the optimization step is upper bounded by it. This is why we now use the autoencoder as a pre-processing step which gives us approximately 91.3% clustering purity. Our future work is to make the pipeline more robust w.r.t. the representation learning step. One way to achieve this may be to keep track of the previous (wrong) labels and the new (correct) ones in the image label inference, and use this information during the training of the classifier or as feedback to prior steps.

As of now, the batch size in the optimization step is fixed to 100. This may not be the best choice for larger combinations, say $(w = 12, h = 12)$, and the optimizer may not find the correct result. We thus plan to dynamically compute the batches of examples (and their size) to increase the performance. Furthermore, adding more inference rules or heuristics to the image label inference step — currently it relies on one simple rule — would help achieve higher image label accuracy for more cases.

In our experiments, similar to [6, 5], we have used the minimum number of training examples for every $w \times h$ combination such that the MNIST training dataset (60K images) is used entirely where every image appears exactly once across all examples. However, we have preliminary results that show that for small $w \times h$ combinations, increasing the number of training examples (i.e., allowing images to appear more than once in the training dataset) leads to higher classification accuracy. For instance, for $h = 2, w = 5$, with a minimum number of examples ($n = 6000$), we achieve 93.05% classification accuracy, where as increasing $n$ to 18000 achieves 98% accuracy. This is not surprising since having an image appear more than once gives the image label inference algorithm a higher chance of correctly resolving its label.

In this work, we have presented a scalable technique for weak-supervised learning using domain constraints instead of labels, and evaluated it on the MNIST dataset with the addition constraint. We have shown that our approach scales better than most recent approaches with respect to both width and height of training examples while the total training time is independent of the width and height. Our approach is applicable to problems with numerical and logical constraints where the input can be clustered and multiple examples can be used together to efficiently evaluate the constraints. To achieve a more general implementation, we plan to tackle the handwritten formula problem as defined in [14] as our next use case. Furthermore, to illustrate the combination of both numerical and logical constraints, we also plan to tackle the Sudoku problem as defined in [5].

## Acknowledgments and Disclosure of Funding

We thank our colleagues Kamalika Das and Jiaxin Zhang from Intuit AI Research Center for discussions and comments.

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
