# OpenReview forum: "A Scalable Technique for Weak-Supervised Learning with Domain Constraints"
_NeurIPS.cc/2022/Workshop/HITY — HITY Workshop NeurIPS 2022_

### Official Review · Reviewer_rZ9r · 2022-10-06
**The paper describes a straightforward approach to solve the MNIST addition problem**

**Rating:** 0
**Confidence:** 3

**Review:**

The paper describes a straightforward approach to solve the MNIST addition problem. In contrast to recent approaches, the authors claim that their approach scales well when the complexity of the problem increases.

Problems:
- The proposed solution is tested on the MNIST addition problem only, whereas [1,2] tested their approach on five different problems. In addition, [1] reported results for competing approaches and stated uncertainty intervals, which this work did not.
- Since a novel approach for a specific task was introduced, the paper does rather not fit this workshop centered around "best practices for faster neural network training".
- The approach is of limited novelty and insights since well-known basic concepts were combined.
- The text is often not clear.

[1] Winters, Thomas, et al. "Deepstochlog: Neural stochastic logic programming." Proceedings of the AAAI Conference on Artificial Intelligence. Vol. 36. No. 9. 2022.

[2] Manhaeve, Robin, et al. "Deepproblog: Neural probabilistic logic programming." Advances in Neural Information Processing Systems 31 (2018).

---

### Official Review · Reviewer_YUeD · 2022-10-14

**Rating:** 1
**Confidence:** 2

**Review:**

**Summary:** This paper proposes a pipeline for image classification that allows
incorporating symbolic domain knowledge. The approach is applied to an MNIST
variant where handwritten digits are supposed to be classified given their
(weighted) sum.

**Strengths, Weaknesses & Questions:**
- In general, the paper is well-written with a clear and thoughtful structure.
The contributions are clear and put into context by referencing relevant related
work.
- Figure 3, line 84-91: The pseudo-code in Figure 3 and the corresponding
descriptions in line 84-91 are not understandable to me. A lot of the components
(like `Ex`, `Unres`, `getUnres`, `resolve` and `img`) are not mentioned in the
text at all. If I understand correctly, this is the step, where the symbolic
constraint is incorporated but I don't understand where exactly and how it is
used.
- Line 92-93: I don't understand why this step is needed. In step 3, every image
is assigned a label. Isn't that the ultimate goal of the pipeline?
- Footnote on page 3: I think it is *crutial* to explain how exactly the
autoencoder is trained since it is used for all $w \times h$ runs. If I
understand correctly, the difficulty of the problem is dependent on $w$ and $h$.
Putting it very negatively: If you make it "easy" for the autoencoder, e.g. by
using $w=h=1$, the good results for the "hard" problems (large $w$ and $h$)
might be simply due to the well-working auto-encoder/clustering.
- Figure 4: I think it would be interesting to isolate the impact of
incorporating the symbolic constraint on the classification accuracy. In step 3,
each image is assigned the label of its cluster. What happens if you skip the
subsequent step (using the constraints to detect and fix wrong labels) and
directly go to step 4? This way you could show what the effect of your *image
label improvement algorithm* (Figure 3) is.
- Line 149-150: You claim that your approach scales better than most recent
approaches in terms of training time. However, you do not provide the training
time for any other approach. I'd suggest adding a baseline to your experiments: a
commonly used method from the literature.

**Minor:**
- Line 18-19: If I understand correctly, you claim in this section that
incorporating domain knowledge as symbolic constraints is especially useful when
constraints are too complex to be learned by an ML model. I'm a bit skeptical
about that statement since e.g. neural networks are so successful because they
*can* learn structure that would be hard to put into a symbolic expression.
- Line 22: What exactly does *high scalability* refer to? Is it the amount of
data that has to be processed, the complexity of the task or something else?
- Line 60: You mention here that the runtime of your approach is linear.
According to Figure 4, it is *constant*.
- Line 67: I think, the formula is not correct: For $w=2$, $j \in \{1, 2\}$
yields the exponents $w-1-j = 0$ and $-1$ (where it should be $1$ and $0$).

---

### Official Review · Reviewer_MxHx · 2022-10-17
**An interesting take on weak-supervised problems**

**Rating:** 1
**Confidence:** 3

**Review:**

The paper proposes a novel method for weak-supervised learning where the relation between training targets and unknown labels is described by a symbolic domain constraint. In particular, the example the paper uses is to learn an MNIST classifier, via targets that are sums of MNIST digits. The proposed method essentially consist of 3 steps. First, a latent representation of the MNIST images is learned as an autoencoder in an unsupervised manner. Then, K-means (with K=#number of MNIST classes) is applied to the latent representation yielding K clusters. Then, for batches of images with known sums, the clusters are assigned an MNIST class each. The predicted class per image is then used to train a supervised CNN. The claim of the paper is that their step-wise method scales better in the constraint complexity at the expense of some final classification accuracy which is an interesting trade-off.  The reduction in combinatorial complexity is achieved via the learned latent representation.

Minor comment: I enjoyed the insightful discussion part of the paper. It would still be interesting to report the final classification accuracy that is mentioned between lines 109-114, even it is not on par with the competitor. Further, I would be interested in a comment on the central assumption that clusters can indeed be assigned a label. For MNIST, the assumption might be approximately true, but for more complex datasets, it is unclear if the clusters learned in an unsupervised way will in any way relate to the ground truth classes. This is a generally known issue that seems to be quite relevant for the proposed method.

---

### Decision · Program_Chairs · 2022-10-20

Accept